# Deep learning approach to describe and classify fungi microscopic images

**Bartosz Zieliński**[1,2]                                          BARTOSZ.ZIELINSKI@UJ.EDU.PL
**Agnieszka Sroka-Oleksiak**[3,4]                                   AGNIESZKA.SROKA@UJ.EDU.PL
**Dawid Rymarczyk**[1,2]                                            DAWID.RYMARCZYK@II.UJ.EDU.PL
**Adam Piekarczyk**[1]                                             ADAM.JAN.PIEKARCZYK@GMAIL.COM
**Monika Brzychczy-Włoch**[4]                                       M.BRZYCHCZY-WLOCH@UJ.EDU.PL

[1] *Faculty of Mathematics and Computer Science, Jagiellonian University, 6 Łojasiewicza Street, 30-348 Kraków, Poland*

[2] *Ardigen, 76 Podole Street, 30-394 Kraków, Poland*

[3] *Department of Mycology, Chair of Microbiology, Faculty of Medicine, Jagiellonian University Medical College, 18 Czysta Street, 31-121 Kraków, Poland*

[4] *Department of Molecular Medical Microbiology, Chair of Microbiology, Faculty of Medicine, Jagiellonian University Medical College, 18 Czysta Street, 31-121 Kraków, Poland*

## Abstract

Preliminary diagnosis of fungal infections can rely on microscopic examination. However, in many cases, it does not allow unambiguous identification of the species by microbiologist due to their visual similarity. Therefore, it is usually necessary to use additional biochemical tests. That involves additional costs and extends the identification process up to 10 days. Such a delay in the implementation of targeted therapy may be grave in consequence as the mortality rate for immunosuppressed patients is high. In this paper, we apply a machine learning approach based on deep neural networks and Fisher vector (advanced bag-of-words method) to classify microscopic images of various fungi species. Our approach has the potential to make the last stage of biochemical identification redundant, shortening the identification process by 2-3 days, and reducing the cost of the diagnosis.

**Keywords:** mycological diagnosis, microscopic images, deep learning, Fisher vector.

## 1. Introduction

Yeast and yeast-like fungi are a component of natural human microbiota (Maiken, 2013). However, as opportunistic pathogens, they can cause surface and systemic infections (Rodrigues et al., 2014). The standard procedure in mycological diagnosis begins with collecting various types of test materials. Then, they are cultured on special media (depending on the sample, with or without prior cultivation), and treated with various biochemical tests. As a result, the entire diagnostic process from the moment of culture to species identification can last 4-10 days (see Figure 1).

In this paper, we apply a machine learning approach based on deep neural networks and Fisher vector (advanced bag-of-words method) to classify microscopic images of various species of fungi. The initial results suggest that this methodology has the potential to replace the last stage of biochemical identification. Hence the appropriate antifungal drug

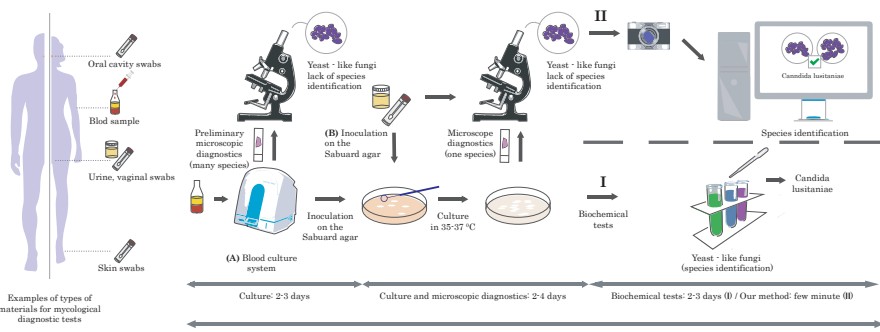

Figure 1: Standard mycological diagnostics (I) require analysis with biochemical resulting in 4-10 days long diagnostic process. In our approach (II), biochemical tests are replaced with a machine learning approach that predicts fungi species based on microscopic images shortening the diagnosis by 2-3 days.

can be introduced 2-3 days earlier. According to our best knowledge, there are no other methods for classifying fungi species based only on microscopic images. Existing methods involve, e.g., biochemical tests, and are costly (Ferrer et al., 2001; Lakner et al., 2012; Papagianni, 2014; Raja et al., 2017).

It is important to stress that except accelerating the mycological diagnosis, the paper aims at presenting the advantages of the approach based on deep learning and Fisher vector over the standard approaches. For this purpose, we analyze the performance and explainability of different models.

## 2. Materials and Method

**Materials.** Our database consists of five yeast-like fungal strains: *Candida albicans* ATCC 10231 (CA), *Candida glabrata* ATCC 15545 (CG), *Candida tropicalis* ATCC 1369 (CT), *Candida parapsilosis* ATCC 34136 (CP), and *Candida lustianiae* ATCC 42720 (CL); two yeast strains: *Saccharomyces cerevisae* ATCC 4098 (SC) and *Saccharomyces boulardii* ATCC 74012 (SB); and two strains belonging to the Basidiomycetes: *Maalasezia furfur* ATCC 14521 (MF) and *Cryptococcus neoformans* ATCC 204092 (CN). All strains are from the American Type Culture Collection. The species in our database highly overlap with the most common fungal infections (Silveira and Husain, 2007; Rodrigues et al., 2014; Lam et al., 2016); however, they are not identical due to the limitations of our repository. Altogether, it contains 180 images (9 strains $\times$ 2 preparations $\times$ 10 images) of resolution $3600 \times 5760 \times 3$ with 16-bits intensity range in every pixel. Images were taken using an Olympus BX43 microscope with 100 times super apochromatic objective under oil-immersion. The photographic documentation was produced with Olympus BP74 camera and CellSense software (Olympus).

**Method.** We consider two types of domain adaptation, both based on DNN features initially pre-trained on an ImageNet (Deng et al., 2009). As a baseline method, we fine-tune

the classifier's block of the well-known network architectures, AlexNet (Krizhevsky et al., 2012), InceptionV3 (Szegedy et al., 2016), and ResNet18 (He et al., 2016), with frozen features' block. All models are trained for 100 epochs using Adam optimizer (Kingma and Ba, 2014) with learning rate $10^{-2}$ (reduced on plateau by factor 2), $\beta_1 = 0.9$, and $\beta_2 = 0.999$. We use cross-entropy as a loss function and apply augmentation to the training data (random rotations, random Gaussian noise, and random scaling between 0.8 to 1.2). As we present in Table 1, they are not optimal; hence, we propose to apply the deep Fisher vector multi-step algorithm (Cimpoi et al., 2015). In contrast to baseline methods, which utilize "shallow" neural network to previously calculated features, our strategies aggregate those features using the Fisher vector (Perronnin and Dance, 2007), and then classify them with support vector machine.

## 3. Experimental setup and results

For the experiments, we split our Digital Images of Fungus Species (DIFaS) database (9 strains × 2 preparations × 10 images) into two subsets, so that both of them contain images of all strains, but from different preparation. Otherwise, the classifier could end up learning clinically irrelevant parameters of the preparation instead of relevant fungi features. We first classify patches instead of the whole image and then aggregate patch-based scores to obtain a scan-based classification. For each fold, we optimize model parameters using internal 5-fold cross-validation. The number of training patches overlapped by less than 50% oscillates between 2000 and 3000, depending on the fold (the number of patches significantly varies depending on the strains).

Table 1: Test accuracy for patch-based and scan-based classification.

| Method | Patch-based | Scan-based |
|---|---|---|
| AlexNet | $71.6 \pm 2.4$ | $77.3 \pm 4.2$ |
| InceptionV3 | $69.9 \pm 1.9$ | $65.9 \pm 4.9$ |
| ResNet18 | $75.9 \pm 2.6$ | $78.3 \pm 5.4$ |
| Fisher vector with AlexNet | $\mathbf{82.4 \pm 0.2}$ | $\mathbf{93.9 \pm 3.9}$ |
| Fisher vector with InceptionV3 | $41.3 \pm 1.9$ | $55.0 \pm 5.6$ |
| Fisher vector with ResNet18 | $71.3 \pm 1.5$ | $88.3 \pm 2.7$ |

As presented in Table 1, Fisher vector with AlexNet works better than all the other baseline methods. It is expected, as a "shallow" neural network (in this case applied to previously calculated features) is less accurate than the Fisher vector (Sánchez and Perronnin, 2011). More surprisingly, the Fisher vector works better with AlexNet than with more advanced architectures. In our opinion, it can be caused by the fact that the features of the latter are more biased on a specific task and therefore do not generalize. It was further confirmed by using T-distributed Stochastic Neighbor Embedding (t-SNE) on features produced by the convolutional block of all considered architectures. We observed that only AlexNet is able to create a representation with a clear separation between the classes.

Finally, we analyze the explainability of the model by visualizing the Gauss centers of the Fisher vector approach using patches closest to the centroids (see Figure 2). Then,

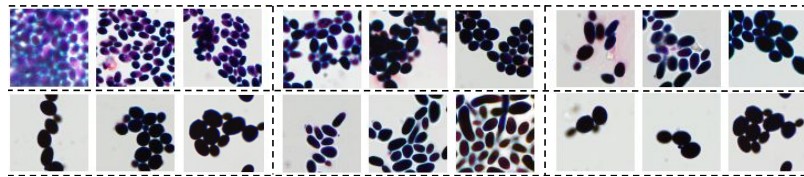

Figure 2: Visualization of six random clusters generated by the Fisher vector (using patches closest to the Gaussian centers).

we describe those clusters using the following set of parameters pre-defined by the microbiologist: brightness (dark or bright), size (small, medium or large), shape (circular, oval, longitudinal or variform), arrangement (regular or irregular), appearance (singular, grouped or fragmentary), color (pink, purple, blue or black), and quantity (low, medium or high). Finally, we examine how the visual information about the main clusters of specific species corresponds to the knowledge of a microbiologist and conclude that it highly correlates. E.g., it revealed that *Candida tropicalis* and *Saccharomyces cerevisiae* have the same main clusters, what confirms their high morphological similarity described in (Krzyciak et al., 2010), i.e., size $3.0 - 8.0 \times 5.0 - 10$ µm, oval shape, elongated, and occurring singly or in small groups). As a result, the internal model inference can be explained to the expert, which builds trust in the system.

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
