# OpenReview forum: "Deep learning approach to describe and classify fungi microscopic images"
_MIDL.io/2020/Conference — MIDL 2020_

### Official Review · AnonReviewer2 · 2020-03-08
**Good application with missing details in the approach**

**Rating:** 2
**Confidence:** 4

**Review:**

This is certainly a very useful application of ML. But many details are missing. Authors talk about Fisher and SVM but this is not visible in the Table 1. The combination of BoVW with deep nets is not explained either. Besides, showing the clusters (Figure 2) hardly qualifies as an explanation for the results.

---

### Official Review · AnonReviewer4 · 2020-03-11
**Well described methods, good evaluation of results but missing some details.**

**Rating:** 3
**Confidence:** 4

**Review:**

In this work, the authors present a machine learning method to classify microscopy images of different fungi strains. The main motivation is to shorten the time of standard mycological diagnostics from 4-10 days to 2-7 days.

The main motivation and methodology are well described. The authors provide a comparison of different approaches and use cross-validation for the optimization of the parameters, which makes the obtained results more reliable. It is important to note the effort of the authors in using a bag of words approach which supports an explanation of what is happening inside the model. While it is not always possible, this approach facilitates the use of machine learning techniques in real applications.

I would like to highlight some points that might be important to get a final version of the text:

- I would consider changing the title to " Deep learning approach to describe and classify fungi microscopic images".
- In the last part of the clinical process described in Figure 1, (Species identification), it is said "99% of identity results". I would not write this as it can be confusing and it is not the accuracy of the method highlighted in Table 1.
- The data is divided into training and validation according to the preparations. Are these 2 preparations independent for each strain and always the same?
- The authors say "split our DIFaS database". Could you please describe the acronym?
- Could it be possible to specify the microscopy modality or set up employed in the image acquisition?
- I miss a short discussion about the real benefits of this approach. In the abstract, it is said that "... microscopic examination ... does not allow unambiguous identification of the species...". Hence, I would like to ask the authors what is the real scope of this kind of approach, especially against a chemical test.  Could it be possible to remove the chemical test from the clinical pipeline? Which are the main drawbacks of this method or what would be necessary to incorporate them into the clinical pipeline?
- In Figure 1 there are two steps in which a microscopic inspection is performed. Could it be possible to work on the classification of the fungi using the images from this preliminary diagnosis?
- Figure 2 contains some characteristic examples of the six clusters obtained from the bag-of-words. Could it be possible to add a label for each of the clusters? Also, it is said, "it revealed that Candida tropicalis and Saccharomyces cerevisiae have the same main clusters". Could you please identify which are the strains that belong to each cluster? Or what are the histograms of each of the strains?

---

### Official Review · AnonReviewer1 · 2020-03-12
**o	In this paper, authors use a deep neural network to classify microscopic fungal images. They intend that their technique be used to cut costs and time of identification in cases of fungal infections. A combined bag-of-words and AlexNet approach achieved the best accuracy (82.4 path-based and 93.9 scan-based) on their testing set.**

**Rating:** 2
**Confidence:** 4

**Review:**

One particular strength of this paper was the authors’ choice of fungal types to train and test their models on. They specify that the chosen types overlap a decent amount with most common fungal infections. Another strength of their paper was the combination of multiple methods, AlexNet and bag-of-words to achieve a better performance than any one method alone.

What are the consequences of misclassifying a fungal infection and treating it with the wrong drug? It seems like, while potentially a useful tool to aid doctors, DNN image classification would not yet be safe to use alone in identifying an infection.
The papers state that “Candida tropicalis” and “Saccharomyces cerevisae” have the same main clusters. Would the image classification system not make similar mistakes in misclassifying similar looking images?

It is unclear that this is a viable solution.  It was stated that  Preliminary diagnosis of fungal infections can rely on microscopic examination. However, in many cases, it does not allow unambiguous identification of the species due to their visual similarity.  If visual similarity is not sufficient, how can one use the images alone to do classification?
There is such a wide variation in deep learning methods - why is this?
more information is necessary to understand the bag of words methods

---

### Official Review · AnonReviewer3 · 2020-03-14

**Rating:** 3
**Confidence:** 3

**Review:**

This paper proposed a few deep learning models to solve the fungi image classification problem. As stated in the paper, it is the first paper that focuses on using image classification help the diagnosis of fungal infections. The paper lacks detailed information about the implementation and model training process, which is very important to draw the conclusion mentioned in the paper.

pro: This paper proposes to use deep learning for fungi microscopic images classification. The problem is interesting and impactful.
The authors provide visualization of random clusters generate by the bag-of-word approach, as well as analysis from a microbiologist perspective.

cons: There are several key points missing in the paper.
1) How the patch are generated from the fungi images and how many total number are used for training?
2) How is each model trained? Without a detailed training setting, it is hard to understand why inceptionV3 performs worse than AlexNet, or why the bag-of-words with InceptionV3 performs much worse than InceptionV3 itself.

---

### Meta-Review · Area_Chair1 · 2020-03-28
**MetaReview of Paper53 by AreaChair1**

**Rating:** 3

**Metareview:**

The reviewers disagree about this paper.  It appears that the motivation is well described and considered to be important by the reviewers.  Another positive aspect of the paper is the choice of fungal types to be characterised by the model, which seem to replicate well the distribution of common fungal infections.  However some reviewers identified lack of details about the implementation and the model training process. In particular, how are the patches generated from the fungi images and how many patches are used for training?  Also, it is surprising that AlexNet performs better than InceptionV3 and that bag-of-words with InceptionV3 performs much worse than InceptionV3.  Another issue identified by the reviewers is that the proposed model seems to be unable to provide an unambiguous species identification because of their visual similarity, which can be potentially devastating for the viability of the method.  The paper has pros and cons, but given the usefulness of the application and the presented results, I am leaning towards acceptance.


**Paper Type:**

validation/application paper

---

### Decision · Program_Chairs · 2020-04-11

Accept